# Vitamin B_12_ in Relation to Oxidative Stress: A Systematic Review

**DOI:** 10.3390/nu11020482

**Published:** 2019-02-25

**Authors:** Erik E. van de Lagemaat, Lisette C.P.G.M. de Groot, Ellen G.H.M. van den Heuvel

**Affiliations:** 1Division of Human Nutrition and Health, Wageningen University and Research, P.O. Box 17, 6700 AA Wageningen, The Netherlands; erikvandelagemaat@gmail.com (E.E.v.d.L.); lisette.degroot@wur.nl (L.C.P.G.M.d.G.); 2FrieslandCampina, Stationsplein 4, 3818 LE Amersfoort, The Netherlands

**Keywords:** B12, cobalamin, oxidative stress, subclinical deficiency, deficiency, triage theory, age-related diseases, micronutrients, ROS, antioxidant

## Abstract

The triage theory posits that modest micronutrient deficiencies may induce reallocation of nutrients to processes necessary for immediate survival at the expense of long-term health. Neglected processes could in time contribute to the onset of age-related diseases, in which oxidative stress is believed to be a major factor. Vitamin B_12_ (B12) appears to possess antioxidant properties. This review aims to summarise the potential antioxidant mechanisms of B12 and investigate B12 status in relation to oxidative stress markers. A systematic query-based search of PubMed was performed to identify eligible publications. The potential antioxidant properties of B12 include: (1) direct scavenging of reactive oxygen species (ROS), particularly superoxide; (2) indirect stimulation of ROS scavenging by preservation of glutathione; (3) modulation of cytokine and growth factor production to offer protection from immune response-induced oxidative stress; (4) reduction of homocysteine-induced oxidative stress; and (5) reduction of oxidative stress caused by advanced glycation end products. Some evidence appears to suggest that lower B12 status is related to increased pro-oxidant and decreased antioxidant status, both overall and for subclinically deficient individuals compared to those with normal B12 status. However, there is a lack of randomised controlled trials and prospective studies focusing specifically on the relation between B12 and oxidative stress in humans, resulting in a low strength of evidence. Further work is warranted.

## 1. Introduction

Vitamin B_12_ (B12), or cobalamin, is an essential water-soluble vitamin that is crucial in maintaining neuronal health and haematopoiesis [1]. Clinical B12 deficiency is rare in developed countries and is primarily caused by genetic aberrations [1]. It often leads to myeloneuropathy or megaloblastic anaemia [1]. However, the potential consequences of asymptomatic subclinical B12 deficiency, thus far, remain unclear. The triage theory posits that asymptomatic subclinical micronutrient deficiencies may induce reallocation of nutrients to processes necessary for immediate survival, at the expense of long-term health [2,3]. Over time, neglected processes could contribute to age-related disease development, in which oxidative stress is believed to be a major factor [4,5]. B12 may possess antioxidant properties [6], and subclinical B12 deficiency may thus contribute to oxidative stress and the onset of age-related diseases. To provide mechanistic context, we first discuss subclinical B12 deficiency, oxidative stress, and the potential antioxidant properties of B12. The subsequent systematic analysis aims to investigate whether lower B12 status is related to increased oxidative stress with specific focus on subclinical B12 deficiency.

### 1.1. Subclinical B12 Deficiency

Clinical deficiency of B12 is relatively easily diagnosed due to its severe symptoms [1]. In line with the triage theory, it is hypothesised that a subclinical B12 deficiency, tentatively defined as 119–200 pmol/L of serum B12 [1,7], could induce long-term damage to macromolecules such as nucleic acids, proteins, and lipids while individuals remain apparently symptom-free [2]. Current prevalence estimates for subclinical deficiency range from 10–15% among individuals >60 years to 23–35% among individuals >80 years [1]. Since hepatic B12 stores vastly exceed the daily loss of the vitamin, deficiencies can remain clinically unexpressed for years [8]. In the absence of genetic causes, subclinical B12 deficiency can be caused by three general factors: inadequate intake, increased demand, and malabsorption [9].

The recommended dietary allowance (RDA) for B12 is 2.4 µg/day for adult males and females [10]. Although this requirement is usually satisfied by an omnivore diet with B12-containing animal-derived products [11], the current RDA may be insufficient to prevent subclinical B12 deficiency; according to Bor et al. [12], 6 µg/day may be more adequate for the general population. It follows that subclinical B12 deficiency warrants further investigation, especially when the potential consequences are considered.

A recent metabolomic analysis found connections between subclinical B12 deficiency and serum metabolic markers of decreased myelin stability, peripheral neuron function, mitochondrial function, and increased oxidative stress [13]. Specifically, the authors posit that the observed correlation between B12 status and acylcarnitines might be related to improved mitochondrial function, which could be relevant for maintaining redox homeostasis [13].

Subclinical B12 deficiency has also been implicated in many age-related diseases that are related to these functions and connections, such as schizophrenia, type 2 diabetes (T2D), Alzheimer’s disease, and Parkinson’s disease [14,15,16,17]. Notably, such age-related diseases share a commonality; oxidative stress is believed to be an important factor in their pathophysiology [5,13,18].

### 1.2. Oxidative Stress and Biomarkers

Oxidative stress occurs when the presence of pro-oxidant compounds such as ROS exceeds the available antioxidant buffering capacity [19,20]. Eukaryotic cells continuously quench free radicals via endogenous antioxidants [21]. It is hypothesised that the ubiquitously expressed nuclear factor-erythroid-2-related factor (Nrf2) is a master regulator of such endogenous antioxidant defences [22,23,24]. Factors such as smoking, infections, and exposure to some chemicals or drugs can also contribute to overall oxidative stress [25,26]. Conversely, exogenous antioxidants such as selenium, vitamin C and E, flavonoids, coumarins, and possibly B12, can aid in reducing it [26].

If oxidative stress occurs, the surplus of ROS can firstly promote inflammation and concomitant cytokine production [18], which in turn can further promote ROS production [27]. Inflammation normally mediates tissue repair in response to aggressors, but it can have detrimental effects if it persists longer than necessary [28]. Low-grade inflammation is believed to be a causative agent in several age-related diseases, including Alzheimer’s and Parkinson’s [28,29]. 

Secondly, ROS can damage functional compounds and tissues through modification of carbohydrates, proteins, lipids, and DNA [26,30]. Carbohydrate oxidation can occur when a carbon-centred radical is formed after ROS bind to the carbohydrate, that upon interaction with other carbohydrates can cause cell-death [26]. Alternatively, it has been suggested that monosaccharide autoxidation plays a role in the pathophysiology of type 2 diabetes [31]. For proteins, amino acid pendant chains can be targeted by ROS to cause peptide fragmentation and the subsequent formation of protein carbonyls [18]. Due to their central role in body functions, dysfunctional proteins can elicit a plethora of deleterious effects; notably, protein oxidation may be related to age-related neurodegeneration [32]. Lipid peroxidation by ROS is a relatively fast-acting consequence of oxidative stress, mostly due to the positive feedback mechanism that accompanies it: the lipid bilayer in cellular membranes can rapidly be disrupted, which allows ROS and other pro-oxidants to easily move through cells [26]. The predominant downstream products of lipid peroxidation are the mutagenic and carcinogenic malondialdehyde (MDA) and 4-hydroxy-2-nonenal (4-HNE) [26,33]. Oxidised low-density lipoproteins (LDL) are suggested to contribute to the development of cardiovascular diseases (CVD) such as atherosclerosis [34]. Finally, ROS can react with and modify DNA, resulting in transcriptional arrest, induction or replication errors, or genomic instability [5,20,26]. The primary targets are sugar and base moieties, oxidation of which leads to DNA cross-linking [25]. One of the primary downstream products of DNA oxidation is the carcinogenic 8-hydroxy-2-deoxyguanosine (8-OHdG) [35]. In vitro experiments with human cell lines suggest that extracellular oxidised DNA decreases Nrf2 activity and may enhance carcinogenesis [36].

These potential targets for ROS encompass most functional compounds, illustrating their broad destructive potential regarding age-related disease development.

### 1.3. Antioxidant Properties of B12

In vitro evidence in human aortic endothelial cells showed that supplementation of physiologically relevant concentrations of cyanocobalamin (a B12 form commonly used in supplements) decreases superoxide levels in the cytosol and the mitochondria [37], though the mechanism of action can be debated. In vitro experiments in cell-free systems and neuronal cells corroborated these findings [38]. Similar results were also reported in vivo in Long-Evan rats: superoxide bursts in retinal ganglion cells were significantly reduced by B12 administration, resulting in increased cell survival [38]. The authors suggest that the enzymatically processed B12 acts as a direct superoxide scavenger [37,38].

In addition, B12 may indirectly stimulate ROS scavenging by preservation of glutathione, which likely involves an intricate network of reactions that has not been fully elucidated [6,39].

B12 might also play a role in modulating immune responses: an associative study in Alzheimer patients found significantly higher basal interleukin-6 production in patients with low B12 status compared to those with normal B12 status [40]. Studies with B12-deficient rats and severely B12-deficient patients also showed increased tumour necrosis factor alpha and decreased epidermal growth factor levels compared to controls [41]. These results suggest that B12 might protect against (low-grade) inflammation-induced oxidative stress by modulating the expression of cytokines and growth factors [41]. It is hypothesised that B12 might achieve this by modifying the activity of transcription factors such as nuclear factor-κB [41].

Folate, vitamin B6, and B12 are important cofactors in homocysteine (Hcy) metabolism [1]. Subclinical B12 deficiency reduces the conversion of Hcy to methionine and thus contributes to intracellular Hcy elevation [1]. Hcy is believed to mediate ROS accumulation through multiple mechanisms, e.g. Hcy auto-oxidation leading to production of H_2_O_2_, which are described in detail elsewhere [42,43].

Finally, if subclinical B12 deficiency indeed mediates oxidative stress through any of the proposed mechanisms, oxidative by-products may in turn impair cellular B12 uptake [44]; this theory concerns advanced glycation end products (AGEs), which constitute proteins or lipids that are glycated by sugar molecules [45]. It has been posited that AGEs can induce oxidative stress and impair cellular B12 uptake [44] and that oxidative stress can likewise contribute to AGE formation [46]. This would create a positive feedback cycle where (subclinical) B12 deficiency mediates oxidative stress, and oxidative stress impairs cellular B12 uptake through AGE formation, thus perpetuating the B12 deficiency [44]. An overview of the potential role of subclinical B12 deficiency in oxidative stress and the onset of age-related diseases is shown in Figure 1.

Subclinical B12 deficiency, tentatively defined as 119–200 pmol/L serum B12 [1,7], may thus contribute to oxidative stress and the onset of age-related diseases. However, whether lower B12 status is quantitatively related to increased oxidative stress markers remains unclear and will be the focus for the remainder of this review.

## 2. Materials and Methods 

This systematic review was conducted following the PRISMA guidelines.

### 2.1. Identification of Records

A systematic query-based PubMed search was performed to identify eligible records that quantitatively measured B12 status markers and oxidative stress markers. The following query was used:

((“oxidative stress” (Mesh)) OR oxidative OR oxidation OR “oxidative stress” OR redox OR tac OR “total antioxidant capacity” OR glutathion* OR gsh OR “glutathion disulphide” OR “glutathione disulphide” OR gssg OR radical* OR malondialdehyde OR mda OR isoprostanes OR f2-isoprostanes OR “breath hydrocarbons” OR carbonyl* OR “8-ohdg” OR “8-hydroxy-2’-deoxyguanosine”) AND ((“vitamin b12” (Mesh)) OR cobalamin OR b12 OR “b-12” OR mma OR “methylmalonic acid” OR holo-tc OR holotc OR holotranscobalamin OR “holo-transcobalamin” OR “holo transcobalamin”) AND (defici* OR shortage OR subclinical OR insufficien* OR scarc* OR inadequa* OR lack)

Searches using the keywords oxidative stress, vitamin B12, and cobalamin were performed in Scopus and Google Scholar to identify additional publications. Reference lists from relevant reviews obtained through the search strategy were also searched. In total, 515 unique records were identified on 8 November 2017 (Figure 2).

### 2.2. In/Exclusion Criteria

All study types and populations were eligible for inclusion due to the expected scarcity and heterogeneity of publications. The following criteria needed to be fulfilled for a study to be included in this review: (1) published ≤10 years ago; (2) in case of human studies, participants had to be adults (≥18 years); (3) at least one quantitative B12 assessment consisting of serum B12, methylmalonic acid (MMA), or holo-transcobalamin (holo-TC); (4) at least one quantitative assessment of oxidative stress markers.

If identified records met any of the following criteria, it was excluded: (1) pregnancy in test subject or focus on offspring; (2) genetic cobalamin disorders in test subjects; (3) manuscript not available in English or Dutch; (4) review publication.

### 2.3. Screening and Eligibility

Records were first selected on relevant publication dates by one researcher, after which 295 were excluded. Title screening was performed by one researcher to exclude clearly irrelevant studies and excluded 95 records. Screening on abstract was performed by three researchers and excluded 49 records. Full-text screening of the remaining 76 records was initially performed by one researcher, and any records where eligibility was uncertain were discussed with a second researcher. In total, 61 records were excluded. A total of 15 records were included; these comprised 13 human trials and 2 animal trials. Reference lists from excluded reviews were searched for additional records and yielded no unique records that met the inclusion criteria.

### 2.4. Data Extraction

A predetermined table was used to perform data extraction and included the following information: author(s), study design, country, description of primary goal and outcome, sample size, age, gender, B12 markers, and oxidative stress markers. This was initially performed by one researcher through full-text reading and discussed with a second researcher until a consensus was reached.

### 2.5. Quality Assessment

Due to the limited and heterogeneous body of evidence that included animal studies, human intervention studies, and observational human studies, risk of bias assessment for individual studies was not performed. The overall body of evidence was narratively discussed in terms of quality and strength of evidence.

### 2.6. Data Analysis

Due to study heterogeneity, a meta-analysis could not be performed. Data were presented in descriptive graphical representations where possible and the remainder was discussed narratively.

The following criteria were used to classify results from individual studies. They were deemed in favour of the antioxidant properties of B12 if: (1) serum B12 or (multiple) other B12 markers differed significantly between study groups, and (2) if ≥50% of the performed statistical tests for oxidative stress markers indicated a significant reduction in pro-oxidant markers or increase in antioxidant markers at a higher B12 status. Studies were deemed against the antioxidant properties of B12 if: (1) serum B12 or (multiple) other B12 markers differed significantly between study groups, and (2) statistical tests for oxidative stress markers showed no indication of improvement for higher B12 status. Studies were deemed ‘unclear’ if: (1) B12 status did not differ significantly between study groups, or (2) non-specific interventions were used that could obscure the potential effect of B12, or (3) >50% of statistical tests for oxidative stress did not reach significance, but showed an overall trend towards improvement, or (4) ≥50% of statistical tests for oxidative stress were significant, but at least one was significant against the oxidant properties of B12.

## 3. Results

### 3.1. Study Characteristics

A total of 15 eligible publications were identified, consisting of 13 human trials and two animal trials. Study characteristics are shown in Table 1.

Human intervention research regarding B12 in relation to oxidative stress is scarce and the strength of evidence is low; with two randomised controlled trials (RCT) with broad-spectrum micronutrient interventions, two retrospective studies (RS), two cross-sectional studies (CS), and seven case-control studies (CC), observational research is predominant. Total sample sizes ranged from 100 [47] to 158 [48] for RCTs, and from 24 [49] to 236 [50] for observational studies. Studies were conducted in a wide range of countries (Table 1). The overall age of study groups ranged from mean 42.4 ± 15.2 years [48] to median 52 years (Q1–Q3 44–56 years) [47] for RCTs, and from mean 28.4 ± 8.6 years [51] to ≥70 years [52] for observational studies.

The human populations under investigation showed a wide range of (disease) phenotypes. Boanca et al. [51] compared lacto-ovo vegetarians with omnivores and three studies investigated diabetic patients [47,53,54]. Other disease phenotypes under investigation in CC designs included cardiovascular disease [55], schizophrenia [56], and chronic myeloproliferative disorder [57]. Otherwise healthy B12-deficient patients were investigated in five studies, consisting of two CC [58,59], one CS [49], and two RS studies [50,52]. Finally, the RCT by Muss et al. [48] focused solely on healthy volunteers.

Eleven out of thirteen human studies measured serum B12. Although assessment of Hcy was common as well, a lack of consensus on the preferred medium was observed: six measured serum Hcy, and six measured plasma Hcy. Serum MMA measurements were performed by three studies, and urine MMA analysis was performed by two. Finally, serum holo-TC was measured by one study.

Commonly measured markers for oxidative stress included MDA, total antioxidant capacity/status (TAC/TAS), glutathione (GSH), and SOD. MDA was measured in a total of seven studies, TAC/TAS in six, GSH in four, and SOD in four. Most studies measured these markers either fluorometrically or colorimetrically in plasma or serum, and many used standardised kits from different manufacturers. Other studied markers included ROS, ox-LDL, glutathione peroxidase (GPx), NADPH oxidase (NOx), catalase, nitrates and nitrites, oxidant risks, and protein carbonyls.

Both animal studies relied on animals with induced B12 deficiency that were compared with non-deficient control animals, namely, mice [60] and worms [61].

### 3.2. Overall B12 Status in Relation to Oxidative Stress

Including all disease phenotypes, species, study types, and other population characteristics, nine out of 15 studies supported the antioxidant properties of B12 (60%), one did not (6.7%), and five showed unclear results (33.3%) (Figure 3A). Five CC studies, one CS, one RS, and two animal studies supported B12 as an antioxidant. Notably, the potential antioxidant effect of B12 was unclear for both RCTs due to their broad-spectrum micronutrient interventions.

Between-group statistical tests performed by the original authors of animal, CC, RCT, and CS studies for the most common oxidative stress markers were also considered separately (Figure 3B). For CS studies, the between-group tests relied on stratification based on B12 status. Of all statistical tests, 75.9% were significant (*p* < 0.05) in favour of the antioxidant properties of B12: 81.8% for MDA; 80.0% for GSH; 85.7% for TAC/TAS; and 62.5% for SOD. Interestingly, when focusing solely on the CC studies, very consistent results were found; 92.9% showed significance for lower B12 status being related to higher oxidative stress.

In addition, associative evidence from the two CS studies largely supported the antioxidant properties of B12 through several significant correlations between B12 status and markers such as catalase, TAC, and MDA [49,54]. Finally, the RS studies by Solomon [50,52] were limited to documenting oxidant risks. He found that mean MMA values were significantly higher (*p* < 0.05) in older patients (≥70 years) with one oxidant risk compared to younger participants (<70 years) with two or more oxidant risks. In addition, when three or more oxidant risks were present, elevated MMA values occurred in 84% of community-dwelling adults.

### 3.3. Subclinical B12 Deficiency in Relation to Oxidative Stress

Subclinically deficient serum B12 values, tentatively defined as 119–200 pmol/L serum B12, were observed in five CC studies (Table 2). All serum B12 measurements were converted to pmol/L using a molecular weight of 1355.388 g/mol if other units were provided [62]. If the median, range, and sample size were reported, mean ± SD were estimated according to Hozo et al. [63]. Three studies supported the antioxidant properties of B12 [53,55,59], one does not [58], and one presented unclear results [51]. Due to heterogeneity in the study populations and oxidative stress markers, grouping of absolute results was not possible. Rather, the percentage difference was calculated to provide tentative quantitative insights based on the relative differences between groups in anti/pro-oxidant markers measures in individual studies. The following formula was used:
|meansubclinical−meannormal|(meansubclinical+meannormal)2×100%=percentage difference

Table 2 shows that ten out of eleven between-group differences for antioxidant markers (90.9%) and four out of six for pro-oxidant markers (66.7%) were statistically significant in support of B12 as antioxidant, for a total of fourteen out of eighteen (77.8%). However, it should be noted that two CC studies [53,55] provided the bulk of evidence in favour of B12 as an antioxidant for subclinically deficient serum B12. Finally, in contrast to human studies that show higher oxidative stress in subclinically deficient compared to normal B12 status, the animal study by Ghosh et al. [60] suggests that severe, but not subclinical, B12 deficiency induces oxidative stress and reduces antioxidant capacity in mice.

## 4. Discussion

This review aimed to investigate whether lower B12 status is related to increased oxidative stress. A total of fifteen recent studies were identified, comprising thirteen human studies and two animal studies. Overall, three main findings can be highlighted:

First, some evidence suggests that lower B12 status is indeed related to increased pro-oxidant and decreased antioxidant status, both overall and for subclinically deficient B12 status compared to normal B12 status. Second, RCTs and prospective studies that focus specifically on the relation between B12 and oxidative stress in healthy humans are lacking, thereby decreasing the overall strength of evidence. Third, a lack of methodological consensus on B12 status and oxidative stress assessment precludes meta-analyses and subsequent quantitative recommendations.

### 4.1. B12 Status in Relation to Oxidative Stress

Overall, the antioxidant properties of B12 are supported by most studies (60.0%). The role of B12 as an antioxidant still holds upon isolation of human studies that compared a subclinically B12 deficient group with a normal B12 status group (60.0%) (Table 2). In addition, >75% of between-group statistical tests for the most common markers (i.e. MDA, GSH, TAC/TAS, and MDA) showed significantly higher oxidative stress when B12 status was lower, both overall and for subclinically deficient serum B12 values compared to normal (Figure 3B). However, the results regarding subclinically deficient serum B12 values largely depend on two individual studies, indicating that the body of evidence is lacking to underscore the triage theory.

Nevertheless, the body of evidence overall tentatively supports the role of B12 as a modulator of redox homeostasis. In apparently asymptomatic individuals with subclinical B12 deficiency, covert oxidative damage may thus occur over time that could contribute to the development of age-related diseases [5,8,15]. A recent in vitro study provides additional mechanistic evidence for the antioxidant properties of B12: Rzepka et al. [64] found that human epidermal melanocytes treated with a synthesised B12 antagonist showed a 120% increase in ROS production after 24 days of incubation compared to control melanocytes. The authors hypothesise that the increased oxidative stress was a result of Hcy accumulation due to methionine synthase inhibition [64]. However, more research is needed to validate these findings.

Animal studies were included in this review to provide additional insights from more controlled settings to complement the results from human studies. Interestingly, the results from human studies regarding subclinical B12 deficiency partially contradict the findings of the animal trial by Ghosh et al. [60], who reported that severe, but not modest, B12 deficiency, induces oxidative stress in mice. It has been posited that microbiotal B12 synthesis regulates B12 status in rodents such as mice in addition to their occasional consumption of animal-based foods [65]. Although Ghosh et al. successfully developed a mouse model of B12 deficiency, such physiological differences may limit extrapolation of these findings to humans, who are entirely dependent on nutritional intake of B12 [60,65].

Several studies reported unclear results regarding B12 status and oxidative stress. Both included RCTs utilised broad-spectrum micronutrient interventions that rendered the isolated effect of B12 supplementation unclear [47,48]. Although both monitored serum B12, their intervention included folate, which is directly related to B12 metabolism [1]. Both studies did find an overall oxidative stress-reducing effect of micronutrient supplementation [47,48], supporting the general principles of the triage theory. Two observational studies showed conflicting results compared to the overall trend. Boanca et al. [51] found in their CC study that lacto-ovo vegetarians had significantly lower serum B12, MDA, and SOD compared to non-vegetarians. Concomitant lower MDA, a product of lipid peroxidation, and SOD, a superoxide scavenger, appears paradoxical as it indicates lower downstream oxidative damage and lower antioxidant capacity [19]. Although mechanistic explanations for the conflicting observations by Boanca et al. remain unclear [51], they may underline the complex nature of oxidative stress [21]. Finally, Güney et al. [58] showed no difference in oxidative stress markers between B12 deficient patients and healthy volunteers. Also, their one-month B12-treatment did not significantly alter markers in B12-deficient patients despite a significant increase in serum B12 [58]. The authors hypothesise that such markers may take longer to improve, which could explain the non-significant findings [58].

Overall, the accumulated evidence suggests that lower B12 status is indeed related to higher oxidative stress. However, these results should be interpreted with caution, as there are several limitations to the body of evidence.

### 4.2. Limitations

Although total serum B12 is certainly the most widely applied marker to assess B12 status, novel markers, such as MMA and holo-TC, have been introduced as better indicators of functional B12 status [1,7]. However, most researchers still primarily opt for serum B12 assessment to quantify B12 status, perhaps due to financial or practical considerations [66]. It has been shown that serum B12 values can be high despite intracellular depletion [7], indicating functional deficiency. Although such deficiency is most often caused by genetic afflictions, it can also occur in otherwise healthy individuals [7]. This may partially explain the large variation found in serum B12 assessment in some included studies [58,59]. Fedosov et al. [67] acknowledge that even holo-TC and MMA are likely not completely independent markers of B12 status and propose a combined indicator, called 4cB12, that includes serum B12, Hcy, holo-TC, and MMA. The primary advantage of this proposed combined marker is that adjustments for age and folate status can be made if necessary, and that the equations can be modified to provide a single indicator even if only two or three markers are measured [1,67]. Theoretically, such a combined marker could circumvent some of the difficulties regarding interstudy comparability, but its clinical and investigative utility remain to be seen.

For oxidative stress, a consensus on appropriate biomarkers is wanting. Due to the complex nature of this condition, biomarkers can range from markers of redox homeostasis to specific downstream tissue damage products [19]. Ideally, multiple markers covering all facets would be combined, rather than focusing on a specific reaction. However, such analytical procedures are time-consuming and expensive [45] and are not often employed in practice. In addition, multiple assays and standardised kits are available for purchase for the same marker, limiting external validity of results even if the same markers are measured [68]. A consensus on the appropriate test medium is lacking as well, further limiting between-study comparisons. In this review, a meta-analysis was impossible due to methodological heterogeneity of biomarkers related to B12 status and oxidative stress.

Furthermore, the triage theory focuses on the prevention of age-related diseases by optimising micronutrient intake [2], which concerns healthy populations. Disease phenotypes such as T2D and schizophrenia are associated with increased oxidative stress [17,69] and do not necessarily translate to healthy individuals. Although studies focusing solely on healthy individuals were scarce, serum B12-deficient participants and vegetarians were considered healthy as well. Seven such studies were included, of which three supported increased oxidative stress related to lower B12 status, three presented unclear results, and one did not support it. The relation between B12 status and oxidative stress in healthy populations thus remains ambiguous.

Finally, human evidence predominantly consists of observational research without prospective cohort designs. As a result, a causal relationship between B12 and oxidative stress cannot be shown, despite the consistent but low-strength results from both human and animal trials pointing towards B12 as an antioxidant.

## 5. Conclusions

Evidence from studies on B12 status in relation to oxidative stress consistently suggests that lower B12 status is related to increased pro-oxidants and decreased antioxidants, both overall and for subclinical B12 deficiency compared to normal B12 status. However, due to a lack of prospective research, consensus on appropriate biomarkers, and interventions focusing specifically on B12, causality cannot be established.

It is therefore imperative that, in the future, agreed-upon gold standard assessments for B12 status and oxidative stress are established. In addition, prospective cohort studies and RCTs in healthy individuals with specific focus on B12 and oxidative stress are warranted to establish causality. If the triage theory proves true, the incidence of age-related diseases could potentially be lowered significantly by adequate B12 intake.

## Figures and Tables

**Figure 1 nutrients-11-00482-f001:**
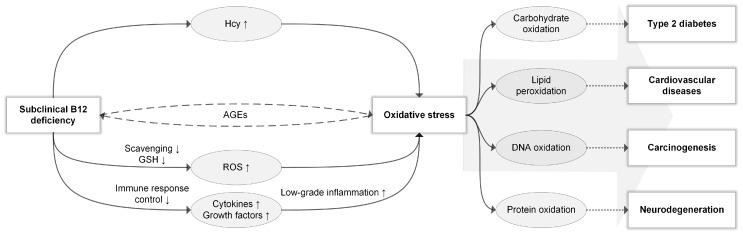
Subclinical B12 deficiency in relation to oxidative stress. Subclinical B12 deficiency might induce oxidative stress through: Hcy accumulation; reduction of ROS scavenging and indirect reduction of GSH; reduction of immune response control by modulating cytokine and growth factor expression with subsequent low-grade inflammation; a positive feedback cycle through advanced glycation end-products (AGEs), where B12 deficiency induces AGE formation through oxidative stress and AGEs reduce B12 uptake. Oxidative stress-induced carbohydrate oxidation has been implicated in type 2 diabetes, lipid peroxidation in cardiovascular diseases, DNA oxidation in carcinogenesis, and protein oxidation in neurodegeneration. However, such associations likely constitute only part of the complicated role of oxidative stress in the pathophysiology of age-related diseases. As indicated by the large arrow, different oxidation substrates are likely synergistically responsible.

**Figure 2 nutrients-11-00482-f002:**
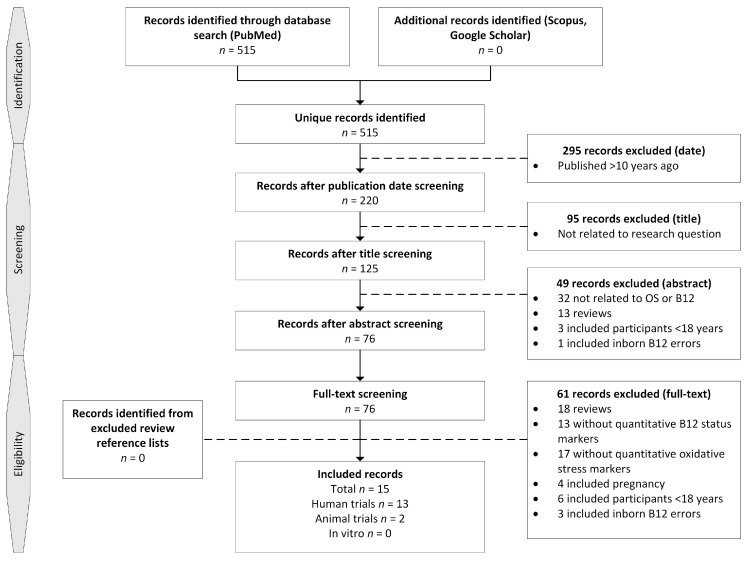
Study selection process.

**Figure 3 nutrients-11-00482-f003:**
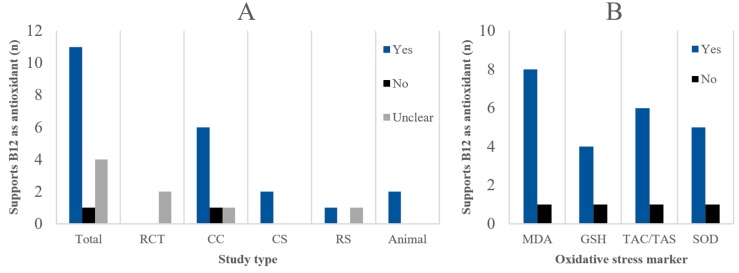
(**A**) Number of studies that overall support, do not support, or show unclear results regarding the antioxidant properties of B12, in total and per study type. Specific criteria for ‘yes’, ‘no’, or ‘unclear’ classification can be found in the methods section. (**B**) Pooled number of statistical tests for all study types for common oxidative stress biomarkers that significantly (*p* < 0.05) support increased oxidative stress or reduced antioxidant capacity in case of lower B12 status.

**Table 1 nutrients-11-00482-t001:** Characteristics of included studies.

Author	Design	Country	Primary Goal	Study Groups (Mean Age ± SD, % Male)	B12 Markers	Oxidative Stress Markers	Supports Antioxidant Properties of B12
Misra et al. 2017 [59]	CC Human	India	Assess oxidative stress markers in B12-deficient patients compared to healthy volunteers and examine correlation among included markers.	51 serum B12 deficient patients (45.78 ± 2.19 years, 70.6% male) 53 healthy volunteers (44.28 ± 2.20 years, 77.8% male)	Serum B12 Serum Hcy	Plasma GSH Serum TAC Plasma MDA	Yes
Waly et al. 2016 [55]	CC Human	Egypt	Evaluate B-vitamins in relation to hyperhomocysteinemia and oxidative stress in cardiac patients compared to healthy volunteers.	25 cardiac patients (54.72 ± 10.3 years, 48.0% male) 25 healthy volunteers (53.81 ± 6.8 years, gender not specified)	Serum B12 Serum Hcy	Serum GSH Serum TAC Serum NN Serum MDA	Yes
Boanca et al. 2014 [51]	CC Human	Romania	Evaluate the impact of the lacto-ovo diet on oxidative stress compared to non-vegetarians regarding B12 status.	48 lacto-ovo vegetarians (28.4 ± 8.6 years, 33.3% male) 38 non-vegetarian volunteers (29.8 ± 1 0.1 years, 34.2% male)	Serum B12	Erythrocyte SOD Serum MDA	Unclear
Özcan et al. 2008 [56]	CC Human	Turkey	Investigate the relationship between B12 status and cell membrane composition regarding oxidative stress, cholesterol, and phospholipid content in schizophrenic patients compared to healthy volunteers.	18 schizophrenic patients (31 ± 7 years, 55.6% male) 20 healthy volunteers (30 ± 8 years, 50.0% male)	Serum B12 Urine MMA Plasma Hcy	Membrane MDA	Yes
Vener et al. 2010 [57]	CC Human	Italy	Determine if folate and/or B12 depletion can lead to hyperhomocysteinemia and contribute to oxidative stress in chronic myeloproliferative disorders compared to healthy volunteers.	51 CMPD patients (median 64 years (range 40–84 years), 52.9% male) 53 healthy volunteers (median 50 years (range 30–84 years), 64.2% male)	Serum B12 Serum holo-TC Serum Hcy	Serum TAC Serum ROS	Yes
Al-Maskari et al. 2012 [53]	CC Human	Oman	Evaluate the status of folate and B12 in relation to serum Hcy and oxidative stress indices in T2D patients compared to healthy volunteers.	50 T2D patients (51.43 ± 7.9 years, 50% male) 50 healthy controls (48.94 ± 6.02 years, 50% male)	Serum B12 Serum Hcy	Serum GSH Serum TAS Serum catalase Serum GPx Serum SOD	Yes
Güney et al. 2015 [58]	CC with one-armed intervention Human	Turkey	Determine the effect of B12 deficiency on oxidative stress compared to healthy controls, and to determine the effect of 1-month cyano-Cbl treatment for B12 deficient patients on oxidative stress (without placebo or control).	40 B12 deficient patients (43.1 ± 1 5.9 years, 25% male) 40 healthy controls (40.1 ± 16.9 years, 42.5% male)	Serum B12 Serum MMA Urine MMA Plasma Hcy	Plasma TOS Plasma TAS OSI	No
Lee et al. 2016 [54]	CS Human	Taiwan	Investigate the correlation between vitamin B12 status and oxidative stress and inflammation in diabetic vegetarians and omnivores.	54 T2D vegetarians (65.1 ± 11.3 years, 30% male) 100 T2D omnivores (57.5 ± 10.5 years, 45% male)	Serum B12	Plasma MDA Serum Ox-LDL-C Erythrocyte SOD Erythrocyte GPx Erythrocyte catalase	Unclear
Hunaiti et al. 2016 [49]	CS Human	Jordan	Assess the impact of B12 deficiency on lipid peroxidation and antioxidant capacity in patients with symptoms and signs of B12 deficiency.	24 patients with B12 deficiency (mean 53 years, range 36–76 years, 37.5% male)	Serum B12 Serum Hcy	Serum MDA Serum TAC	Yes
Gariballa et al. 2013 [47]	RCT Human	UAE	Test the effect of 3 months B-vitamins and antioxidant supplementation compared to placebo on antioxidant capacity and oxidative stress in obese T2D patients.	50 T2D patients allocated to intervention (median 52 years, Q1–Q3 44–56 years, 46% male) 50 T2D patients allocated to placebo (median 51 years, Q1–Q3 42–60 years, 36% male)	Serum B12 Plasma Hcy	Protein carbonyl Plasma MDA Plasma GSH	Unclear
Muss et al. 2015 [48]	RCT Human	Austria	Test the neuroprotective effect of verum supplementation (measured at 3 and 6 months) compared to placebo with a focus on oxidative stress in healthy volunteers.	116 volunteers allocated to intervention (42.4 ± 15.2 years, 39.7% male) 43 volunteers allocated to placebo (45.8 ± 15.5 years, 25.6% male)	Serum B12 Serum Hcy	Serum FORD Serum FORT Serum lipid peroxidation Serum SOD	Unclear
Solomon 2015 [52]	RS Human	USA	Explore the association between functional B12 deficiency and oxidative stress in elderly and younger patients.	170 community-dwelling adults with serum B12 ≥ 400 pg/mL were retrospectively reviewed and stratified <70 years (*n* = 100, 42% male) and ≥70 years (*n* = 70, 54.3% male)	Serum MMA Plasma Hcy	Oxidant risks	Yes
Solomon 2016 [50]	RS Human	USA	Examine the relationship between MMA/Hcy and oxidant risks in community-dwelling adults with low (≤200 pg/mL) and low-normal (201-300 pg/mL) serum B12.	49 participants with low serum B12 (57 ± 19 years, 29% male) 187 participants with low-normal serum B12 (56 ± 17 years, 39% male)	Serum MMA Plasma Hcy	Oxidant risks	Unclear
Ghosh et al. 2016 [60]	Animal	Japan	Evaluate differential effects of severe and moderate B12 deficiency on several factors, including oxidative stress.	10 severely B12 deficient mice 10 moderately deficient mice 10 control mice	Plasma Hcy	Liver MDA Liver protein carbonyl Liver SOD Liver catalase	Yes
Bito et al. (2017) [61]	Animal	India	Clarify levels of oxidative stress and induced damage when B12 deficiency is present using a *C. elegans* animal model. In addition, the relationship between B12 deficiency and memory impairment is investigated.	12 B12-deficient worms 12 control worms	Plasma Hcy	H_2_O_2_ NN MDA Protein carbonyls GSH Total SOD Catalase GPx	Yes

CC: case-control; B12: vitamin B_12_; Hcy: homocysteine; GSH: glutathione; TAC: total antioxidant capacity; MDA: malondialdehyde; NN: nitrates and nitrites; SOD: superoxide dismutase; MMA: methylmalonic acid; CMPD: chronic myeloproliferative disorder; holo-TC: holo-transcobalamin; ROS: reactive oxygen species; T2D: type 2 diabetes; TAS: total antioxidant status; GPx: glutathione peroxidase; TOS: total oxidant status; OSI: oxidative stress index; CS: cross-sectional study; Ox-LDL-C: oxidised low-density lipoprotein C; RCT: randomised controlled trial; UAE: United Arab Emirates; FORD: free oxygen radicals defence; FORT: free oxygen radicals test; RS: retrospective study; USA: United States of America.

**Table 2 nutrients-11-00482-t002:** Subclinical B12 deficiency in relation to oxidative stress; results from case-control studies.

	Serum B12 ^A^ (pmol/L, Mean ± SD)			Antioxidant Marker ^B,^* (% Difference between Groups)					Pro-Oxidant Marker ^B,^* (% Difference between Groups)			
Author (Year)	Subclinical Group	Normal Group	Supports B12 as Antioxidant	SOD	GSH	CAT	GPx	TAC TAS	TOS	OSI	NN	MDA
Misra (2017) [59]	172.4 ± 17.8	304.0 ± 213.6	Yes	†	−9.3 *	†	†	−9.0 *	†	†	†	28.8 *
Waly (2016) [55]	154.4 ± 10.9	272.9 ± 23.7	Yes	†	−111.6 *	†	†	−54.9 *	†	†	113.6 *	76.2 *
Boanca (2014) [51]	125.7 ± 44.7	282.5 ± 90.2	Unclear	−6.1 *	†	†	†	†	†	†	†	−16.3 *
Al-Maskari (2012) [53]	186.9 ± 19.0	508.0 ± 30.2	Yes	−70.9 *	−78.3 *	−92.7 *	−66.6 *	−90.9 *	†	†	†	†
Güney (2015) [58]	136.8 ± 40.3	562.3 ± 314.8	No	†	†	†	†	0.0	−8.0	−40.0	†	†

* Indicates a significant difference between groups on this marker as stated by the authors. † Not measured. A: Mean ± SD was estimated according to Hozo et al. if median, range, and sample size were reported [63]. Serum B12 concentrations were calculated in pmol/L using a molecular weight of 1355.388 g/mol if other units were reported [62]. B: Percentage differences were calculated using the standard formula: │(meansubclinical − meannormal)│/((meansubclinical + meannormal)/2) × 100%. Positive and negative values indicate that these markers were respectively increased or decreased in subclinical serum B12 groups compared to normal serum B12 groups. Abbreviations: SD: standard deviation; SOD: superoxide dismutase; GSH: glutathione; CAT: catalase; GPx: glutathione peroxidase; TAC/TAS: total antioxidant capacity/status; TOS: total oxidant status; OSI: oxidative stress index; NN: nitrates and nitrites; MDA: malondialdehyde.

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
