# Peer review of "Vitamin B12 in Relation to Oxidative Stress: A Systematic Review"

_nutrients, 2019, doi:10.3390/nu11020482_

Reviewer 1 Report

The authors conducted a systematic review of existing literature (past 10 years from November 2017) published in PubMed about the association between vitamin B12 and markers of oxidative stress.

The article is difficult to follow and mix a review and systematic review in one. The introduction is very long for a systematic review.

It is not clear if the authors intended to do a metaanalysis from the beginning. In my opinion for performing a metaanalysis there should be a preliminary search and this would end up with no metaanalysis being possible. The topic is very narrow and the outcome detection was very heterogeneous. Thus no collective conclusion can be made based on the search the authors conducted.   

In my opinion it is not appropriate to mix studies in human, mice and worms but on the other hand to exclude studies on children. It is not clear why to limit the search to the last 10 years, if the appropriate studies are anyway limited in number. There are some treatment studies with vitamin B12 (or as a part of multivitamins) that may be taken as starting points: eg. PMID: 19056591    

 Author Response

We thank the reviewers for their critical and detailed review of our manuscript. We believe that the comments, suggestions, and corresponding revisions have enhanced the quality and readability of our manuscript. Below you will find point-by-point responses to the remarks made by the reviewers and the revisions that were made as a result. Reviewer comments are bulleted and shown in bold typeface, our replies in normal typeface, and citations from the manuscript in italic typeface. Revisions in the manuscript are presented in red text.

Reviewer 1

The article is difficult to follow and mix a review and systematic review in one. The introduction is very long for a systematic review.

Our systematic review aimed to quantitatively investigate the relation between vitamin B12 and (markers of) oxidative stress. To our knowledge, the relevant mechanisms that could underpin the potential antioxidant effect of B12 have not been summarised before. As such, we aimed to succinctly review these mechanisms before presenting the results of our systematic analysis. Although this indeed led to a relatively long introduction, we believe that it provides a more complete story and could aid the reader to put the results into context.

The first paragraph of the introduction was altered to better reflect this intention and guide the reader (line 40):

“To provide mechanistic context, we first briefly discuss subclinical B12 deficiency, oxidative stress, and the potential antioxidant properties of B12. The subsequent systematic analysis aims to investigate whether lower B12 status is related to increased oxidative stress with specific focus on subclinical B12 deficiency.”

It is not clear if the authors intended to do a metaanalysis from the beginning. In my opinion for performing a metaanalysis there should be a preliminary search and this would end up with no metaanalysis being possible. The topic is very narrow and the outcome detection was very heterogeneous. Thus no collective conclusion can be made based on the search the authors conducted.

We did intend to perform a meta-analysis from the beginning. Although our preliminary search showed that the body of evidence was limited, a meta-analysis was not deemed impossible until the full-text screening phase. We also believe that this result – insufficient/inadequate data for a meta-analysis – is an important part of the results in itself; it shows that the methodology regarding assessment of B12 status and oxidative stress markers is highly heterogeneous. It is unlikely that strong conclusions on this topic can be formulated until some form of standardisation is incorporated in future studies. Hence, the reviewer is correct that a collective conclusion cannot be drawn from the current review. We highlight this in the discussion and conclusion:

Section 4.2 first discusses the heterogeneity of markers for B12 status and oxidative stress until line 385, which states: “In this review, a meta-analysis was impossible to carry out due to methodological heterogeneity of biomarkers related to B12 status and oxidative stress.”

Next, the lack of established causality is mentioned from line 397 onwards: “As a result, a causal relationship between B12 and oxidative stress cannot be shown, despite the consistent but low-strength results from both human and animal trials pointing towards B12 as an antioxidant.”

The conclusion reiterates these statements starting at line 403.

We hope that these statements will help future experimental study authors decide on their analysis methods and data presentation.

In my opinion it is not appropriate to mix studies in human, mice and worms but on the other hand to exclude studies on children.

Although the review briefly discusses overall results (with human and animal studies combined), we focus specifically on subclinical B12 deficiency in humans in section 3.3. Because this was planned in advance, we wanted to keep human evidence restricted to adults. Children were excluded to negate the additional complications that growth and development would introduce to an already complicated topic. For that reason, publications such as the study mentioned by the reviewer (PMID: 19056591), which included children with autism, were not considered.

Animal studies were included because they can provide (mechanistic) insights from a more controlled setting to complement low-strength human evidence. The results were discussed in relation to the body of human evidence (e.g. from line 336) and evidence regarding subclinical deficiency in humans remained separate. Considering that the review cannot form strong conclusions based on the current state of research, we believe that the animal studies help to provide a more complete picture on the topic.

To communicate this to the reader, the following was added to the discussion (section 4.1, line 335):

“Animal studies were included in this review to provide additional insights from more controlled settings to complement the results from human studies.”

It is not clear why to limit the search to the last 10 years, if the appropriate studies are anyway limited in number.

A range of 10 years was agreed upon in an early phase of the process to keep the review manageable. Because a total of 15 studies were included, we felt that this was a proper number of publications to review and that revision of the range was not necessary. Although the body of evidence is indeed somewhat scarce overall, the most prominent finding was that the quality of evidence was low.

 There are some treatment studies with vitamin B12 (or as a part of multivitamins) that may be taken as starting points: eg. PMID: 19056591

The study mentioned by the reviewer (PMID: 19056591) included children with autism. Due to potential complications that growth and development could introduce to the already heterogeneous study populations, we chose to focus on human studies with adults.

Reviewer 2 Report

General comments.

The manuscript “Vitamin B12 in relation to oxidative stress: A systematic review” by Lagemaat et al concerns an important subject, namely “hidden” activities of cobalamin (Cbl, vitamin B12). A number of papers have been published during the recent two decades, where the antioxidant activities of Cbl (either direct or indirect) were tested, though with no unequivocal conclusion reached. The submitted review summarizes these results and critically analyzes the materials from literature. In this regard, the manuscript might provide a considerable help for many researchers. It should be mentioned on the downside, that the submitted review avoids any mentioning of possible mechanisms, which might explain (at least theoretically) the antioxidant function of Cbl (see Introduction, chapter 1.3). Lack of mechanisms makes the action of B12 rather “mystic” with some resemblance to a medicine man remedy. Yet, the descriptions of several relevant mechanisms exist in the literature, though some publications date back to 1950s – 1970s. The current review, however, concentrates on very recent publications, which sometimes come with rather absurd statements, probably because of lacking knowledge about the early “classical” works, see Main specific comments. A more critical presentation of the literature would help the reader. I suggest a thorough revision of chapter 1.3, where mechanisms of Cbl's antioxidant activity should be given. The overall language of the manuscript is rather awkward, and the attached pdf-file contains numerous suggestions for improvement of the text flow. Major revision is requested.

Main specific comments

Chapter 1.3 discusses three pathways for the antioxidant action of Cbl. No mechanisms are mentioned for any of these pathways, see the below comments for each process taken separately.

Chapter 1.3 (line 102) states “In vitro evidence suggests that B12 actively scavenges ROS, particularly superoxide, at a rate similar to superoxide dismutase (SOD) [37].” Citation of ref. 37 is particularly problematic because it reports “a kinetic study of the reaction between superoxide and cob(II)alamin.” Yet, superoxide (O2-) reacts only with the oxidized from of cob(III)alamin ([CoIII]Cbl), while ordinary oxygen O2, indeed, binds to cob(II)alamin ([CoII]Cbl), see references below. Both reactions lead to temporary formation of O2[CoII]Cbl, which undergoes decomposition according to the scheme:

4×O2[CoII] + 2×H2O→ 4×[CoIII] + 3×O2 + 4×HO-

The aforementioned materials were discussed in refs. [Bayston et al. (1969) Superoxocobalamin: the first intermediate in the autoxidation of vitamin B12r, J. Am. Chem. Soc. 91, 2775–9; Ellis et al. (1973) Formation of a Co–O2 complex by the co-ordination of free superoxide ion. J. Chem. Soc., Chem. Commun., 1973, 0, 781b-782, doi:  10.1039/C3973000781B; and J.M. Pratt, 'Inorganic Chemistry of Vitamin B, Academic Press, London, 1972, chapter 11.V]. The mechanism of superoxide “deactivation” suggested in ref. 37 is apparently a misinterpretation. The authors use catalase in the medium to decompose hydrogen peroxide (erroneously suggested as a byproduct). In reality, presence of catalase would lead to oxidation of superoxide (O2-) without participation of Cbl according to the below scheme, which involves the auto-catalytic dismutation of superoxide in aqueous medium:

O2- + H+ → HOO·;   2×HOO· → H2O2 + O2;    2×H2O2 →(catalase)→ O2 + 2×H2O

The final product of dismutation (O2) will afterward react with [CoII]Cbl and give O2[CoII]Cbl, which generates [CoIII]Cbl as shown above. In the view of all aforementioned, I find citation of ref. 37 very inappropriate.

Chapter 1.3 (line 105) states “In addition to its direct scavenging role, B12 may indirectly stimulate ROS scavenging as well by enhancing methionine synthase activity to form glutathione [6,39].” An explanation is needed, because the readers would remain confused. For example, methionine synthase (MS + B12) catalyzes conversion of homocysteine (Hcy) to methionine and decreases the level Hcy. Yet, Hcy is a precursor in glutathione (GSH) synthesis, and a low activity of MS (at low B12) should (in principle) promote GSH synthesis via a high Hcy. Yet, this is not the case. Thus, a model in ref. 39 does not show any association of a low MS activity (at low B12) with GSH concentration. Quite opposite, many publications show that low B12 leads to low GSH. The mechanism of interaction between B12 and GSH levels is under discussion. For example, it was suggested in literature that a high Hcy causes oxidative stress via a fast self oxidation with the accompanying production of hydrogen peroxide (2Hcy-SH + O2 → Hsy-S-S-Hcy + H2O2), reviewed in ref. [Loscalzo (1996) The oxidant stress of hyperhomocyst(e)inemia. J Clin Invest. 98(1):5-7, doi: 10.1172/JCI118776].  It seems that high Hcy does not stimulate GSH production but degrades to Hsy-S-S-Hcy, and the accompanying generation of H2O2 gradually exhausts GSH stores, involved in decomposition of H2O2.

Chapter 1.3 (line 107) states “... B12 may also modulate cytokine and growth factor production to offer protection from immune response-induced oxidative stress [27,40].” Ref. 27 does not mention B12 in any way, and this quotation is irrelevant within the shown context. Ref 40 does not show any experimental proof or references that B12 modulates cytokine and growth factor production. Therefore, other publications (if any) should be quoted in this regard. In principle, Cbl (most probably as deoxyadenosyl- and methyl-forms) can affect expression of proteins acting through a riboswithch (see e.g. Nahvi et al (2004) Coenzyme B12 riboswitches are widespread genetic control elements in prokaryotes. Nucleic Acids Research, 32, 143–150, doi: 10.1093/nar/gkh167). Yet, I do not recall that such activity was detected in animal cells. Please, specify the sources, which show that B12 modulates cytokines.

Language corrections are given in the attached document.

Author Response

We thank the reviewers for their critical and detailed review of our manuscript. We believe that the comments, suggestions, and corresponding revisions have enhanced the quality and readability of our manuscript. Below you will find point-by-point responses to the remarks made by the reviewers and the revisions that were made as a result. Reviewer comments are bulleted and shown in bold typeface, our replies in normal typeface, and citations from the manuscript in italic typeface. Revisions in the manuscript are presented in red text.

We thank the reviewer for the extensive language suggestions in the attached PDF. A vast majority was accepted and revised and, in our opinion, enhanced the readability of the review. The alterations are shown in red text in the manuscript. Some additional remarks on the suggestions in the PDF:

Regarding the addition of H202 to Hcy in image 1: We purposefully did not go into detail on the mechanisms by which Hcy mediates oxidative stress. We believe that this is a topic that deserves its own research, shown for example by the numerous mechanisms discussed in the review that we refer to in the manuscript (https://doi.org/10.1152/ajpheart.00548.2005). As such, this was not revised. However, the potential role of Hcy auto-oxidation with H2O2 production in Hcy-induced oxidative stress was briefly mentioned in the text (line 126).

Regarding the descriptions of mechanisms related to the antioxidant properties of B12 (section 1.3): The main purpose of section 1.3 was to provide brief context on the proposed antioxidant properties of B12 to complement the quantitative analysis. Major revisions were made to the paragraph based on your suggestions (see the remarks below). However, we believe that a more in-depth discussion on the potential underlying mechanisms warrants a separate review. It is indeed a very complex and interesting field, but in our opinion beyond the scope of the introduction of this manuscript.

Regarding the deletion of the sentence starting at line 385 of the original manuscript (line 403 in the revised version): We agree that this statement was rather broad, it has been deleted. We felt that it was important to relate back to the triage theory, but this was already done in the final sentence (line 409 in the revised manuscript).

Below you will find our point-by-point responses to the main specific comments.

Chapter 1.3(line 102) states “In vitro evidence suggests that B12 actively scavenges ROS, particularly superoxide, at a rate similar to superoxide dismutase (SOD) [37].” Citation of ref. 37 is particularly problematic because it reports “a kinetic study of the reaction between superoxide and cob(II)alamin.” Yet, superoxide (O2-) reacts only with the oxidized from of cob(III)alamin ([CoIII]Cbl), while ordinary oxygen O2, indeed, binds to cob(II)alamin ([CoII]Cbl), see references below.

Although other works have adopted the conclusion of the relevant study with statements similar to ours (e.g. https://doi.org/10.1016/j.ajpath.2017.08.032), the detailed comments made by the reviewer clearly warranted a critical revision of the statement.

Upon further inspection, Moreira et al. appear to have expanded upon this preliminary work with more extensive experiments (PMID: 21672628). Physiologically relevant concentrations of CNCbl were shown to effectively protect against intracellular levels of superoxide in the cytosol and in the mitochondria, independent of Hcy. Catalase was not used in this experiment, which negates the potential misinterpretation of superoxide inactivation in the earlier work as stated by the reviewer. The authors state that these results combined with their previous kinetic study provides suggest that B12 may indeed be a direct scavenger of superoxide. These findings are supported by studies in cell-free systems, neuronal cells, and in vivo in Long-Evan rats.

Reference 37 has been changed to the more extensive work by Moreira et al. (PMID: 21672628) and the following replaces the original section (line 106):

In vitro evidence in human aortic endothelial cells showed that physiologically relevant concentrations of cyanocobalamin, a B12 form commonly used in supplements, protect against intracellular superoxide levels in the cytosol and in the mitochondria [37]. In vitro experiments in cell-free systems and neuronal cells corroborated these findings [38]. Similar results were also reported in vivo in Long-Evan rats: superoxide bursts in retinal ganglion cells were significantly reduced by B12 administration, resulting in increased cell survival [38]. Thus, B12 appears to act as a direct superoxide scavenger [37,38].

Chapter 1.3(line 105) states “In addition to its direct scavenging role, B12 may indirectly stimulate ROS scavenging as well by enhancing methionine synthase activity to form glutathione [6,39].” An explanation is needed, because the readers would remain confused.

We thank the reviewer for the in-depth clarification on this comment. After reviewing the mentioned concerns, we agree that the sentence did not accurately reflect the complex relationship between B12, HCy, and GSH. Most studies indeed show that low B12 status is associated with low GSH, but the mechanism remains a topic of discussion. Although a thorough discussion on this relationship is beyond the scope of this review, a revision was warranted to clarify the statement. The following replaces the relevant sentence (line 113):

“In addition, B12 may indirectly stimulate ROS scavenging by preservation of glutathione, which likely involves an intricate network of reactions that has not been fully elucidated [6,39].”

The abstract was also revised to include this (line 19):

“…indirect stimulation of ROS scavenging by preservation of glutathione…”

The proposed role of Hcy auto-oxidation with H2O2 was briefly mentioned in the section regarding Hcy (line 126):

“Hcy is believed to mediate ROS accumulation through multiple mechanisms, e.g. Hcy auto-oxidation leading to production of H2O2, which are described in detail elsewhere [42].”

Chapter 1.3(line 107) states “... B12 may also modulate cytokine and growth factor production to offer protection from immune response-induced oxidative stress [27,40].” Ref. 27 does not mention B12 in anyway, and this quotation is irrelevant within the shown context.

The reviewer is correct that this paper (PMID: 23508734) does not mention B12, this publication was erroneously cited in this section of the manuscript and the in-text reference has been removed.

Ref40 does not show any experimental proof or references that B12modulates cytokine and growth factor production. Therefore, other publications (if any) should be quoted in this regard.

The relevant reference (PMID: 25459143) states the following: “Cobalamin may protect against oxidative stress via modulation of the immune system, affecting production of cytokines and growth factors”. However, the reviewer is correct that the author neglected to include citations in support of that statement. We agree that this invalidates the initial statement and the aforementioned reference was removed from the manuscript.

An additional publication was identified that presents original associative data regarding B12 status and cytokines in Alzheimer patients (PMID: 20110595). The authors found significantly higher basal IL-6 production in patients with low B12 status (<250 pg/ml) compared to those with normal B12 status. These findings are corroborated by studies that showed increased levels of TNF-α and reduced EGF in B12-deficient rats and severely B12-deficient patients (covered in PMID: 19409980). It is also mentioned that modification of NF-κB activity may be a plausible mechanism for these findings.

Since these findings are more nuanced than the original statement in the manuscript, the sentence has been replaced by the following (line 115):

“B12 might also play a role in modulating immune responses: an associative study in Alzheimer patients found significantly higher basal interleukin-6 production in patients with low B12 status compared to those with normal B12 status [40]. Studies with B12-deficient rats and severely B12-deficient patients also showed increased tumour necrosis factor alpha and decreased epidermal growth factor levels compared to controls [41]. These results suggest that B12 might protect against (low-grade) inflammation-induced oxidative stress by modulating the expression of cytokines and growth factors [41]. It is hypothesised that B12 might achieve this by modifying the activity of transcription factors such as nuclear factor-κB [41].”

Reviewer 3 Report

The Manuscript entitled “VITAMIN B12 IN RELATION TO OXIDATIVE STRESS: A SYSTEMATIC REVIEW” is generally well written and concerns an interesting and up-to date health problem – vitamin B12 subclinical deficiency. However, in my opinion the manuscript requires several improvements:

Line 47: „… could induce long-term damage” – I suggest that Authors should specify this statements by adding  “damage of...”

Line 60: I suggest that Authors should determine the nature of connections by adding brief information about effect of subclinical vitamin B12 deficiency on mitochondrial function.

Line 71: In my opinion “some chemicals and drugs” rather than “chemicals and drugs” should be used by the Authors, because there are also some chemicals and drugs with antioxidative properties.

Line 93-94: The sentence “Finally, ROS can break DNA strands by oxidising nucleic acids, thus inducing genetic disruption” should be rewritten because there is some confusions, e.g. which type of nucleic acids?

Line 111-112: The sentence “Consistent with its metabolism, subclinical B12 deficiency mediates intracellular homocysteine (Hcy) elevation in concert with dietary folate and vitamin B6” should be rewritten – in the current version it may be unclear for the readers

The Authors should include information about in vitro (in cellulo) studies, in which vitamin B12 deficiency was induced in normal human cell lines resulting in cellular oxidative stress [e.g. Rzepka et al. Vitamin B12 deficiency induces imbalance in melanocytes homeostasis - A cellular basis of hypocobalaminemia pigmentary manifestations. Int J Mol Sci. 2018; 19(9)]. These findings can be treated as another confirmation of the hypothesis concerning the role of lower vitamin B12 status in oxidative stress-mediates disorders.

The list of references includes some errors:

-     Item [13] - inconsistency with other items (Capital letters)

-      Item [19] - the authors of cited article are: Czerska, M,;  Mikołajewska, K.; Zieliński, M.; Gromadzińska, J.; Wąsowicz, W.

 Author Response

We thank the reviewers for their critical and detailed review of our manuscript. We believe that the comments, suggestions, and corresponding revisions have enhanced the quality and readability of our manuscript. Below you will find point-by-point responses to the remarks made by the reviewers and the revisions that were made as a result. Reviewer comments are bulleted and shown in bold typeface, our replies in normal typeface, and citations from the manuscript in italic typeface. Revisions in the manuscript are presented in red text.

Line 47: „… could induce long-term damage” – I suggest that Authors should specify this statements by adding  “damage of...”

Agreed, the sentence (line 48) has been altered to specify the affected molecules according to Ames (PMID: 17101959):

“…could induce long-term damage to macromolecules such as nucleic acids, proteins, and lipids…”

Line 60: I suggest that Authors should determine the nature of connections by adding brief information about effect of subclinical vitamin B12 deficiency on mitochondrial function.

To clarify the relevant connection, the following brief explanation was added (line 63):

“Specifically, the authors posit that the observed correlation between B12 status and acylcarnitines might be related to improved mitochondrial function, which could be relevant for maintaining redox homeostasis [13].”

Line 71: In my opinion “some chemicals and drugs” rather than “chemicals and drugs” should be used by the Authors, because there are also some chemicals and drugs with antioxidative properties.

Agreed, the sentence did not reflect that some chemicals and drugs indeed show antioxidative properties. The suggestion was accepted and revised to include this (line 75):

“…some chemicals and drugs…”

Line 93-94: The sentence “Finally, ROS can break DNA strands by oxidising nucleic acids, thus inducing genetic disruption” should be rewritten because there is some confusions, e.g. which type of nucleic acids?

Agreed, the sentence did not clearly communicate the effects on DNA. It has been rewritten as follows (line 97):

“Finally, ROS can react with and modify DNA, resulting in transcriptional arrest, induction or replication errors, or genomic instability [5,20,26]”

The next sentence briefly describes the primary targets (line 99):

“The primary targets are sugar and base moieties, oxidation of which leads to DNA cross-linking [25]”

Line 111-112: The sentence “Consistent with its metabolism, subclinical B12 deficiency mediates intracellular homocysteine (Hcy) elevation in concert with dietary folate and vitamin B6” should be rewritten – in the current version it may be unclear for the readers

The sentence was rewritten as follows to increase clarity (line 123):

“Folate, vitamin B6, and B12 are important cofactors in homocysteine (Hcy) metabolism [1]. Subclinical B12 deficiency reduces the conversion of Hcy to methionine and thus contributes to intracellular Hcy elevation [1].

The Authors should include information about in vitro (in cellulo) studies, in which vitamin B12 deficiency was induced in normal human cell lines resulting in cellular oxidative stress [e.g. Rzepka et al. Vitamin B12 deficiency induces imbalance in melanocytes homeostasis - A cellular basis of hypocobalaminemia pigmentary manifestations. Int J Mol Sci. 2018; 19(9)]. These findings can be treated as another confirmation of the hypothesis concerning the role of lower vitamin B12 status in oxidative stress-mediates disorders.

We thank the reviewer for this valuable suggestion; the mentioned study (PMID: 30235895) was published after we last added new references to the manuscript and we would have missed it if the suggestion was not offered. It indeed provides additional confirmation of the hypothesis. The following was added to the discussion (line 329):

“A recent in vitro study provides additional mechanistic evidence for the antioxidant properties of B12: Rzepka et al. found that human epidermal melanocytes treated with a synthesised B12 antagonist showed a 120% increase in ROS production after 24 days of incubation compared to control melanocytes. The authors hypothesise that the increased oxidative stress was a result of Hcy accumulation due to methionine synthase inhibition. However, more research is needed to validate these findings.”

Item [13] - inconsistency with other items (Capital letters)

Accepted and revised.

Item [19] - the authors of cited article are: Czerska, M,;  Mikołajewska, K.; Zieliński, M.; Gromadzińska, J.; Wąsowicz, W

Accepted and revised.

Round  2

Reviewer 1 Report

i have reviewed the manuscript in my previous report due to lack of merit, no convincing rational behind including animal studies and omitting studies on children, restricting the search to the last 10 years in a field that is not well developed, and other issues related to the design. These points are still not resolved.

Author Response

We again thank the reviewers for their comments. As instructed, no alterations were made to the manuscript based on the remarks by reviewer 1 since no new comments were proposed.

Reviewer 2 Report

General comments.

The revised manuscript presents a sufficient improvement.

Specific comments.

Lines 106 – 108. Change the text to “... that supplementation of physiologically relevant concentrations of cyanocobalamin (a B12 form commonly used in supplements) decreases superoxide levels in the cytosol and the mitochondria [37], though the mechanism of action can be debated.

Lines 11 – 112. Change the text to “The authors suggest that the enzymatically processed B12 acts as a direct superoxide scavenger [37,38].”

Line 327.  Change the text to “In apparently asymptomatic individuals …”, because the original text “... ostensibly asymptomatic individuals …“ literally means that these individuals pretended on purpose to be deficient. Here is an appropriate example of expression “Their ostensible goal was to clean up government corruption, but their real aim was to unseat the government.”

Author Response

We again thank the reviewers for their comments.  Below you will find point-by-point responses to the remarks made by reviewer 2. Reviewer comments are bulleted and shown in bold typeface, our replies in normal typeface, and quotes from the revised manuscript in italic typeface. Revisions in the manuscript are presented in red text.

Reviewer 2

Lines 106 – 108. Change the text to “... that supplementation of physiologically relevant concentrations of cyanocobalamin (a B12 form commonly used in supplements) decreases superoxide levels in the cytosol and the mitochondria [37], though the mechanism of action can be debated.

Accepted and revised, the full sentence now reads (line 106):

“In vitro evidence in human aortic endothelial cells showed that supplementation of physiologically relevant concentrations of cyanocobalamin (a B12 form commonly used in supplements) decreases superoxide levels in the cytosol and the mitochondria [37], though the mechanism of action can be debated.”

Lines 11 – 112. Change the text to “The authors suggest that the enzymatically processed B12 acts as a direct superoxide scavenger [37,38].”

Accepted and revised, the sentence now reads (line 112):

“The authors suggest that the enzymatically processed B12 acts as a direct superoxide scavenger [37,38].”

Line 327.  Change the text to “In apparently asymptomatic individuals …”, because the original text “... ostensibly asymptomatic individuals …“ literally means that these individuals pretended on purpose to be deficient. Here is an appropriate example of expression “Their ostensible goal was to clean up government corruption, but their real aim was to unseat the government.”

Accepted and revised. We thank the reviewer for the clarification, ‘ostensibly’ was indeed an improper use of the word. The full sentence now reads (line 328):

“In apparently asymptomatic individuals with subclinical B12 deficiency, covert oxidative damage may thus occur over time that could contribute to the development of age-related diseases [5,8,15].”